# Domain Generalization by Learning and Removing Domain-specific Features

**Yu Ding**
University of Wollongong
yd624@uowmail.edu.au

**Lei Wang**
University of Wollongong
leiw@uow.edu.au

**Bin Liang**
University of Technology Sydney
Bin.Liang@uts.edu.au

**Shuming Liang**
University of Technology Sydney
Shuming.Liang@uts.edu.au

**Yang Wang**
University of Technology Sydney
Yang.Wang@uts.edu.au

**Fang Chen**
University of Technology Sydney
Fang.Chen@uts.edu.au

## Abstract

Deep Neural Networks (DNNs) suffer from domain shift when the test dataset follows a distribution different from the training dataset. Domain generalization aims to tackle this issue by learning a model that can generalize to unseen domains. In this paper, we propose a new approach that aims to explicitly remove domain-specific features for domain generalization. Following this approach, we propose a novel framework called Learning and Removing Domain-specific features for Generalization (LRDG) that learns a domain-invariant model by tactically removing domain-specific features from the input images. Specifically, we design a classifier to effectively learn the domain-specific features for each source domain, respectively. We then develop an encoder-decoder network to map each input image into a new image space where the learned domain-specific features are removed. With the images output by the encoder-decoder network, another classifier is designed to learn the domain-invariant features to conduct image classification. Extensive experiments demonstrate that our framework achieves superior performance compared with state-of-the-art methods. Code is available at https://github.com/yulearningg/LRDG.

## 1 Introduction

Deep Neural Networks (DNNs) have achieved great performance in computer vision tasks [26]. However, the performance would drop if the test dataset follows a distribution different from the training dataset. This issue is also known as domain shift [39]. Recent research has found that DNNs tend to learn decision rules differently from humans [17, 21, 16]. For example, in ImageNet-based [37] image classification tasks, Convolutional Neural Networks (CNNs) tend to learn local textures to discriminate objects, while we humans could use the knowledge of global object shapes as cues. The features learned by the DNNs may only belong to specific domains and are not generalized for other domains. For example, in real-world photos, objects belonging to the same category have similar textures, but in sketches [27], objects are only drawn by lines and contain no texture information. For a CNN that uses textures to discriminate objects in the photos, poor performance can be expected when it is applied to the sketches. This situation calls for DNNs that can learn features invariant across domains instead of learning features that are domain-specific.

36th Conference on Neural Information Processing Systems (NeurIPS 2022).

In this paper, we focus on the research topic of domain generalization and follow the multiple source domain generalization setting in the literature. Its goal is to train a model that can perform well on unseen domains. In this setting, we can access multiple labeled source domains and one or more unlabeled target domains. All the source and target domains share the same label space. During the training process, the source domains are available but the target domains are unseen. The target domains are only provided in the test phase.

One typical approach to domain generalization is to learn domain-invariant representations across domains [18, 30, 42, 3, 11, 14, 45, 31, 35]. This approach is based on the assumption that each domain has its domain-specific features and that all domains share domain-invariant features. For example, textures are domain-specific features for the photos but shapes are domain-invariant features for both photos and sketches. Previous works propose methods that seek to distill the domain-invariant features. Although demonstrating promising performance, these methods do not clearly inform the deep neural networks that the domain-specific features shall be effectively removed. Instead, it is only hoped that they would be removed through achieving the final goal of learning the domain-invariant features. The lack of this clear guidance to the network may affect its learning efficacy. In this paper, we propose a new approach that aims to explicitly remove the domain-specific features in order to achieve domain generalization. As indicated above, CNNs tend to learn the domain-specific features rather than the domain-invariant features for classification. To prevent this from taking place, we actively remove the domain-specific features and guide the CNNs to learn the domain-invariant features for classification. Following this approach, we propose a novel framework: Learning and Removing Domain-specific features for Generalization (LRDG).

Our framework consists of domain-specific classifiers, an encoder-decoder network, and a domain-invariant classifier. The training process of our framework includes two steps. In the first step, each domain-specific classifier is designed to effectively learn the domain-specific features from one source domain. Specifically, a domain-specific classifier is designed to discriminate the images across different classes within one particular source domain. At the same time, this classifier is required to be unable to discriminate the images across different classes within any other source domain. Each source domain therefore corresponds to one domain-specific classifier under this design. In the second step, the encoder-decoder network maps the input images into a new image space where the domain-specific features learned above are to be removed from the input images by utilizing the domain-specific classifiers. Different from the first step, each domain-specific classifier here is unable to discriminate the mapped images across different classes within the corresponding source domain. The mapped images are expected to contain much fewer domain-specific features compared with the original input images. The domain-invariant classifier is then appended to the encoder-decoder network and trained with the mapped images. By this design, the encoder-decoder network actively removes the domain-specific features and the domain-invariant classifier will be better guided to learn the domain-invariant features. Once trained, the encoder-decoder network and the domain-invariant classifier will be used for the classification of the unseen target domains.

It is worth noting that our framework is different from the data augmentation based methods for domain generalization [43, 34, 46, 7]. Our framework aims to remove the domain-specific features from the input images while the data augmentation based methods generate various images with novel domain-specific features. Besides, our framework just maps the input images into a new image space and does not augment them to enlarge the training dataset.

We demonstrate the effectiveness of our framework with experiments on three benchmarks in domain generalization. Our framework consistently achieves state-of-the-art performance. We also experimentally verify that our framework effectively reduces the distribution difference among the source and target domains according to the generalization risk bound in the literature [2].

## 2 Proposed framework

Assuming that we are given $N$ source domains $\mathcal{D}_s = \{D_s^1, D_s^2, \ldots, D_s^N\}$ which follow different distributions. For each domain (dataset), $D_s^i = \{(\mathbf{x}_j^i, y_j^i)\}_{j=1}^{n_i}$ where $n_i$ is the number of samples in $D_s^i$, and $(\mathbf{x}_j^i, y_j^i)$ is the data-label pair for the $j$th sample in the $i$th domain. Following the literature, we assume that all source and target domains share the same label space. The goal of domain generalization is to use these source domains $\mathcal{D}_s$ to learn a model for the unseen target domain $D_t$.

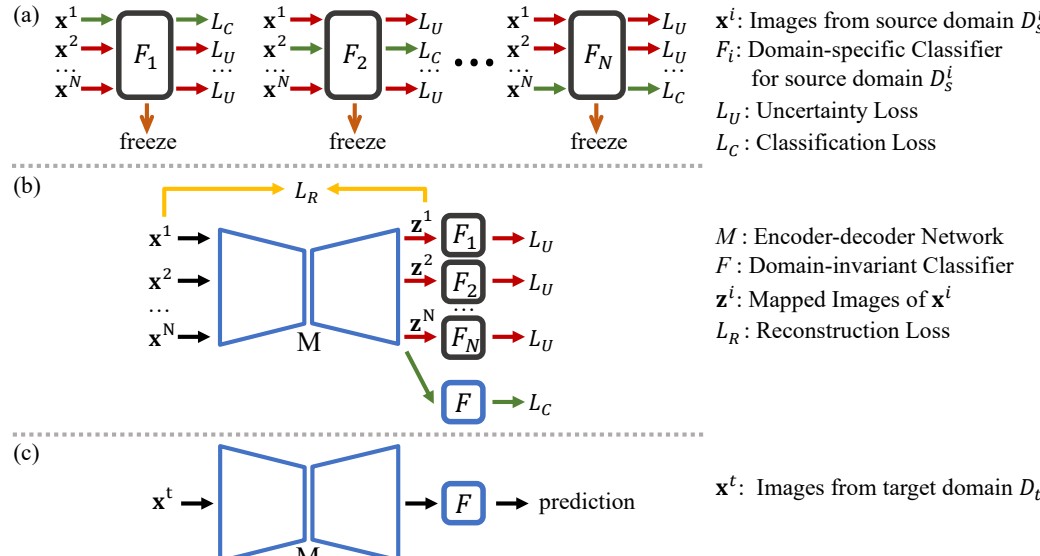

Figure 1: An overview of the proposed framework LRDG. (a) The domain-specific classifier $F_i$ is trained with the classification loss $L_C$ on the source domain $D_s^i$ and the uncertainty loss $L_U$ on the remaining source domains. After training, the weights of all the domain-specific classifiers are frozen. (b) The encoder-decoder network $M$ is trained with the reconstruction loss $L_R$ and the uncertainty loss $L_U$ through the domain-specific classifiers. Meanwhile, the domain-invariant classifier $F$ is trained with the classification loss $L_C$ on the mapped images. (c) In the testing phase, the encoder-decoder network $M$ and the domain-invariant classifier $F$ are used for classification on the target domain $D_t$.

Our work is inspired by recent work [32], where it uses a "lens" network (i.e. image-to-image translation network) to remove "shortcuts" (low-level visual features that a CNN can quickly learn, such as watermarks and color aberrations) from input images in a self-supervised learning task. Differently, our work focuses on removing the domain-specific features from the input images for the domain generalization task. We use an encoder-decoder network similar to the "lens" network, but we design a different method to leverage the encoder-decoder network to remove the domain-specific features. In this section, we illustrate our framework in detail. We also provide theoretical analysis for our framework. Fig. 1 gives an overview of the entire framework.

## 2.1 Learning domain-specific features

Our framework starts by training $N$ individual domain-specific classifiers $\mathcal{F}_S = \{F_1, F_2, \ldots, F_N\}$ in which the classifier $F_i$ is designed to only use the domain-specific features from the source domain $D_s^i$ to discriminate images. The domain-specific classifiers $\mathcal{F}_S$ should not use the domain-invariant features as cues. In other words, $F_i$ is expected to be able to effectively discriminate images across different classes within $D_s^i$ but it should be difficult for $F_i$ to discriminate images across different classes within any other domains. Domains excluding $D_s^i$ are used to maximize the classification uncertainty or adversarially increase the difficulty of classification for $F_i$. The classification performance of $F_i$ on the domains excluding $D_s^i$ should be similar to a random guess.

Specifically, the classifier $F_i$ is trained by minimizing a classification loss $\mathcal{L}_C^{F_S}$ on $D_s^i$,

$$\underset{\theta_i}{\arg\min} \, \mathbb{E}_{D_s^i \sim \mathcal{D}_s}[\mathbb{E}_{(\mathbf{x}_j^i, y_j^i) \sim D_s^i}[L_C(F_i(\mathbf{x}_j^i; \theta_i), y_j^i)]], \tag{1}$$

and maximizing an uncertainty loss $\mathcal{L}_U^{F_S}$ on the remaining domains $\{D_s^1, \ldots, D_s^{i-1}, D_s^{i+1}, \ldots, D_s^N\}$,

$$\underset{\theta_i}{\arg\max} \, \mathbb{E}_{D_s^k \sim \mathcal{D}_s, k \neq i}[\mathbb{E}_{(\mathbf{x}_j^k, y_j^k) \sim D_s^k}[L_U(F_i(\mathbf{x}_j^k; \theta_i))]], \tag{2}$$

where $\theta_i$ denotes the parameters of the classifier $F_i$. $L_C$ and $L_U$ are the classification loss function and the uncertainty loss function, respectively. We use the cross-entropy loss as the classification loss. For the uncertainty loss, since we aim to make the prediction similar to a random guess, we use entropy loss,

$$L_U(F_i(\mathbf{x}_j^k; \theta_i)) = - \sum_{l=1}^{C} p(y = l | F_i(\mathbf{x}_j^k; \theta_i)) \log p(y = l | F_i(\mathbf{x}_j^k; \theta_i)), \tag{3}$$

where $C$ is the number of classes and $p(y = l | F_i(\mathbf{x}_j^k; \theta_i))$ denotes the probability of $\mathbf{x}_j^k$ belonging to class $l$. Least likely loss [32] is an alternative to the entropy loss. The classifier first predicts an image and obtains the probabilities of all the classes. The class with the lowest probability is called the least likely class. This image is assigned with a label of this class. Then we train the classifier to predict the least likely class. The least likely loss is

$$L_U(F_i(\mathbf{x}_j^k; \theta_i)) = L_C(F_i(\mathbf{x}_j^k; \theta_i), \hat{y}_j^k), \; where \; \hat{y}_j^k = \arg\min_y p(y | F_i(\mathbf{x}_j^k; \theta_i)). \tag{4}$$

However, experiments show that the entropy loss can better achieve the classification randomness than the least likely loss, so we use the entropy loss as the default uncertainty loss.

After training, we freeze the parameters $\theta$ of these domain-specific classifiers $\mathcal{F}_S$ and use these classifiers to learn domain-invariant features.

## 2.2 Removing domain-specific features

To remove the domain-specific features learned by the domain-specific classifiers, we utilize an encoder-decoder network $M$ that maps the images into a new image space $\mathcal{Z}$. The output images are fed into the domain-specific classifiers $\mathcal{F}_S$ and a new domain-invariant classifier $F$.

Unlike the training of the domain-specific classifier $F_i$ where the source domain $D_s^i$ is used for minimizing the classification loss, on the contrary, the source domain $D_s^i$ in this step is used to maximize the uncertainty loss $\mathcal{L}_U^M$,

$$\arg\max_{\theta_M} \mathbb{E}_{D_s^i \sim \mathcal{D}_s} [\mathbb{E}_{(\mathbf{x}_j^i, y_j^i) \sim D_s^i} [L_U(F_i(M(\mathbf{x}_j^i; \theta_M); \theta_i))]]. \tag{5}$$

The parameters $\theta_i$ of $F_i$ are frozen and the parameters $\theta_M$ of the encoder-decoder network $M$ are trained. Maximizing the uncertainty loss forces the output image $\mathbf{z}_i = M(\mathbf{x}_i)$ to contain fewer domain-specific features than the input images. In doing so, the encoder-decoder network can remove the domain-specific features in the input images $\mathbf{x}$ and retain domain-invariant features in the output images $\mathbf{z}$.

To maintain the overall similarity between the input and output images, we add a reconstruction loss $\mathcal{L}_R^M$ for the encoder-decoder network,

$$\arg\min_{\theta_M} \mathbb{E}_{D_s^i \sim \mathcal{D}_s} [\mathbb{E}_{(\mathbf{x}_j^i, y_j^i) \sim D_s^i} [L_R(M(\mathbf{x}_j^i; \theta_M), \mathbf{x}_j^i)]], \tag{6}$$

where $L_R$ is the reconstruction loss function. We use pixel-wise $l_2$ loss as the default reconstruction loss for its simplicity and reasonably good performance. Other reconstruction losses could also be employed, such as pixel-wise $l_1$ loss and perceptual losses [24]. Detailed discussion is available in the supplementary material.

We then train the domain-invariant classifier $F$ by minimizing the classification loss $\mathcal{L}_C^{FM}$ on the output images of all the source domains,

$$\arg\min_{\theta_M, \theta_F} \mathbb{E}_{D_s^i \sim \mathcal{D}_s} [\mathbb{E}_{(\mathbf{x}_j^i, y_j^i) \sim D_s^i} [L_C(F(M(\mathbf{x}_j^i; \theta_M); \theta_F), y_j^i)]], \tag{7}$$

where $\theta_F$ are the parameters of the domain-invariant classifier $F$. This classification loss $\mathcal{L}_C^{FM}$ also updates the encoder-decoder network to prevent the encoder-decoder network from losing the domain-invariant features due to the uncertainty loss. The uncertainty loss also has the potential to remove the domain-invariant features if it is difficult to separate the domain-specific features from the domain-invariant features.

Overall, when training the domain-specific classifiers we optimize

$$\mathcal{L}_1 = \mathcal{L}_C^{F_S} + \lambda_1 \mathcal{L}_U^{F_S}, \tag{8}$$

and when learning the domain-invariant features, we optimize

$$\mathcal{L}_2 = \mathcal{L}_C^{FM} + \lambda_2 \mathcal{L}_U^M + \lambda_3 \mathcal{L}_R^M, \tag{9}$$

where $\lambda_1$, $\lambda_2$ and $\lambda_3$ are hyperparameters that control the relative weight of these losses.

For convenience, we denote the encoder-decoder network $M$ and the domain-invariant classifier $F$ as a domain-invariant model. In the testing phase, the domain-invariant model is used for classification on the target domain $D_t$.

## 2.3 Explanation of LRDG with respect to existing theory

We first introduce the generalization risk bound for domain generalization [2] and then further explain the effectiveness of our framework with respect to this.

Theoretically, the corresponding task for a domain is defined as a deterministic true labeling function $f$, where $f : \mathcal{X} \to \mathcal{Y}$. $\mathcal{X}$ and $\mathcal{Y}$ are the input space and the label space, respectively. We denote the space of the candidate hypothesis as $\mathcal{H}$, where a hypothesis $h : \mathcal{X} \to \mathcal{Y}$. The risk of the hypothesis $h$ on a domain $\mathcal{D}$ is defined as

$$\mathcal{R}[h] = \mathbb{E}_{x \sim \mathcal{D}}[\mathcal{L}(h(x) - f(x))], \tag{10}$$

where $\mathcal{L} : \mathcal{Y} \times \mathcal{Y} \to \mathcal{R}_+$ measures the difference between the hypothesis and the true labeling function.

Following [2], for the source domains $\{\mathcal{D}_s^1, \mathcal{D}_s^2, \ldots, \mathcal{D}_s^N\}$, we define the convex hull $\Lambda_S$ of the source domains as a set of mixture source distributions: $\Lambda_S = \{\bar{\mathcal{D}} : \bar{\mathcal{D}}(\cdot) = \sum_{i=1}^N \pi_i \mathcal{D}_s^i(\cdot), 0 \leq \pi_i \leq 1, \sum_{i=1}^N \pi_i = 1\}$. We also define $\bar{\mathcal{D}}_t \in \Lambda_S$ as the closest domain to the target domain $\mathcal{D}_t$. $\bar{\mathcal{D}}_t$ is given by $\arg\min_{\pi_1, \ldots, \pi_N} d_{\mathcal{H}}[\mathcal{D}_t, \sum_{i=1}^N \pi_i \mathcal{D}_s^i]$, where $d_{\mathcal{H}}[\cdot, \cdot]$ is $\mathcal{H}$-divergence [25] that quantifies the distribution difference of two domains. We use the following generalization risk bound [2] for the target domain $\mathcal{D}_t$.

**Theorem 1 (Generalization risk bound [2])** *Given the previous setting, the following inequality holds for the risk $\mathcal{R}_t[h]$, $\forall h \in \mathcal{H}$ for any domain $\mathcal{D}_t$,*

$$\mathcal{R}_t[h] \leq \sum_{i=1}^N \pi_i \mathcal{R}_s^i[h] + \frac{\gamma + \epsilon}{2} + \lambda_\pi, \tag{11}$$

*where $\gamma = d_{\mathcal{H}}[\mathcal{D}_t, \bar{\mathcal{D}}_t]$, $\epsilon = \sup_{i,j \in [N]} d_{\mathcal{H}}[\mathcal{D}_s^i, \mathcal{D}_s^j]$ and $\lambda_\pi$ is the minimum sum of the risks achieved by some $h \in \mathcal{H}$ on $\mathcal{D}_t$ and $\bar{\mathcal{D}}_t$. $\gamma$ measures the distribution difference between the source domains and the target domain. $\epsilon$ is the maximum pairwise $\mathcal{H}$-divergence among source domains.*

Theorem 1 shows that the upper bound for the target domain depends on $\gamma$ and $\epsilon$. We show that our framework could lower the value of this generalization risk bound for a given domain generalization task. Recall that our encoder-decoder network maps the input images into a new image space. We denote the mapped source domains as $\{\widehat{\mathcal{D}}_s^1, \widehat{\mathcal{D}}_s^2, \ldots, \widehat{\mathcal{D}}_s^N\}$ and the mapped target domain as $\widehat{\mathcal{D}}_t$. With the domain-specific classifiers, many domain-specific features are removed from the source domains and the features of the mapped source domains tend to be more domain-invariant. As a result, the mapped source domains $\{\widehat{\mathcal{D}}_s^1, \widehat{\mathcal{D}}_s^2, \ldots, \widehat{\mathcal{D}}_s^N\}$ would have smaller distribution difference than the raw source domains, i.e. $d_{\mathcal{H}}[\widehat{\mathcal{D}}_s^i, \widehat{\mathcal{D}}_s^j] \leq d_{\mathcal{H}}[\mathcal{D}_s^i, \mathcal{D}_s^j]$, indicating that $\epsilon$ in Eq. 11 would probably be reduced. After removing the domain-specific features for each source domain, the mapped target domain $\widehat{\mathcal{D}}_t$ would be closer to the mapped source domains, so our framework could also be likely to reduce $\gamma$ in Eq. 11. Concerning Theorem 1, these changes provide a principled explanation and warrant to the effectiveness of the proposed framework. We will demonstrate these changes in the experiment section (Sec. 3.3).

# 3 Experiments

We evaluate our framework on three benchmark datasets and compare the performance with previous methods. After that, we study the domain divergence among the source and target domains.

### 3.1 Datasets and settings

**Datasets.** We evaluate our framework on three object recognition datasets for domain generalization. PACS [27] contains four domains: Photo (**P**), Art Painting (**A**), Cartoon (**C**) and Sketch (**S**) with each domain covering seven categories including dog, elephant, giraffe, guitar, horse, house, and person. VLCS [39] also has four domains: PASCAL VOC 2007 (**V**), LabelMe (**L**), Caltech (**C**) and Sun (**S**). The images belong to five categories of bird, chair, car, dog, and person. Office-Home [40] has images from 65 categories over four domains including Art (**A**), Clipart (**C**), Product (**P**), and Real-World (**R**). For each dataset, following the literature, the experimental protocol is to consider three domains as the source domains and the remaining one as the target domain.

**Networks and loss functions.** We use U-net [36] for the encoder-decoder network. Following the standard setting in the domain generalization literature [13, 45, 22], we use AlexNet [26], ResNet18 [20] and ResNet50 [20] as backbones for the domain-specific classifiers and the domain-invariant classifier. We use AlexNet for PACS and VLCS, ResNet18 for PACS and Office-Home, and ResNet50 for PACS. AlexNet and ResNet are pre-trained by ImageNet [37] for all the experiments. We use the standard cross-entropy loss as the classification loss $L_C$. For the uncertainty loss $L_U$, we choose the entropy loss. For the reconstruction loss $L_R$, we utilize the pixel-wise $l_2$ loss. A detailed analysis of the loss functions is available in the supplementary material.

**Training setting.** The encoder-decoder network, the domain-specific classifiers, and the domain-invariant classifier are all optimized with Stochastic Gradient Descent. The source datasets are split into a training set and a validation set. The learning rate is decided by the validation set. We set $\lambda_1 = 1$ for all the experiments. We give equal weight to the classification loss and the uncertainty loss for training the domain-specific classifiers. For $\lambda_2$ and $\lambda_3$, we follow the literature [13, 4] and directly use the leave-one-domain-out cross-validation to select their values.

**Methods for comparison.** We compare our framework with previous domain generalization works including domain-invariant based methods [30, 41, 11, 45, 14, 31, 35, 8] and other state-of-the-art methods [15, 4, 9, 28, 13, 46, 34, 22, 7, 44, 10] including data augmentation based methods [34, 46, 7], meta-learning based methods [4, 28, 13], etc. The baseline is defined as the method of empirical risk minimization (ERM). It trains a classifier by minimizing the classification loss on all source domains.

Table 1: Comparison with existing methods on PACS.

| | | | | | AlexNet | | | | | | |
|---|---|---|---|---|---|---|---|---|---|---|---|
| Target | Baseline | CIDDG [30] | JiGen [9] | Epi-FCR [28] | MASF [13] | PAR [41] | DMG [11] | ER [45] | MetaVIB [14] | RSC [22] | LRDG (ours) |
| **A** | 61.13 | 62.70 | 67.63 | 64.70 | 70.35 | 68.70 | 64.65 | 71.34 | 71.94 | 71.62 | **72.01** |
| **C** | 68.77 | 69.73 | 71.71 | 72.30 | 72.46 | 70.50 | 69.88 | 70.29 | 73.17 | **75.11** | 72.97 |
| **P** | 87.96 | 78.65 | 89.00 | 86.10 | 90.68 | 90.40 | 87.31 | 89.92 | **91.93** | 90.88 | 89.50 |
| **S** | 58.63 | 64.45 | 65.18 | 65.00 | 67.33 | 64.60 | 71.42 | 71.15 | 65.94 | 66.62 | **74.86** |
| Avg. | 69.12 | 68.88 | 73.38 | 72.00 | 75.21 | 73.54 | 73.32 | 75.67 | 75.74 | 76.05 | **77.33** |

| | | | | | ResNet18 | | | | | | |
|---|---|---|---|---|---|---|---|---|---|---|---|
| Target | Baseline | Epi-FCR [28] | MASF [13] | DMG [11] | ER [45] | MixStyle [46] | SagNet [34] | Stylized [7] | StableNet [44] | RSC [22] | LRDG (ours) |
| **A** | 77.95 | 82.10 | 80.29 | 76.90 | 80.70 | **84.10** | 83.58 | 82.73 | 81.74 | 83.43 | 81.88 |
| **C** | 74.24 | 77.00 | 77.17 | **80.38** | 76.40 | 78.80 | 77.66 | 77.97 | 79.91 | 80.31 | 80.20 |
| **P** | 95.89 | 93.90 | 94.99 | 93.35 | **96.65** | 96.10 | 95.47 | 94.95 | 96.53 | 95.99 | 95.21 |
| **S** | 70.11 | 73.00 | 71.69 | 75.21 | 71.77 | 75.90 | 76.30 | 81.61 | 80.50 | 80.85 | **84.65** |
| Avg. | 79.54 | 81.50 | 81.03 | 81.46 | 81.38 | 83.70 | 83.25 | 84.32 | 84.69 | 85.15 | **85.48** |

| | | | | | ResNet50 | | | | | | |
|---|---|---|---|---|---|---|---|---|---|---|---|
| Target | Baseline | Metareg [4] | DSON [38] | DMG [11] | ER [45] | RSC [22] | MatchDG [31] | SWAD [10] | Fishr [35] | mDSDI [8] | LRDG (ours) |
| **A** | 82.89 | 87.20 | 87.04 | 82.57 | 87.51 | 87.89 | 85.61 | **89.30** | 88.40 | 87.70 | 86.57 |
| **C** | 80.49 | 79.20 | 80.62 | 78.11 | 79.31 | 82.16 | 82.12 | 83.40 | 78.70 | 80.40 | **85.78** |
| **P** | 95.01 | 97.60 | 95.99 | 94.49 | **98.25** | 97.92 | 97.94 | 97.30 | 97.00 | 98.10 | 95.57 |
| **S** | 72.29 | 70.30 | 82.90 | 78.32 | 76.30 | 83.35 | 78.76 | 82.50 | 77.80 | 78.40 | **86.59** |
| Avg. | 82.67 | 83.60 | 86.64 | 83.37 | 85.34 | 87.83 | 86.11 | 88.10 | 85.50 | 86.20 | **88.63** |

### 3.2 Main results

PACS contains four domains of Art painting, Cartoon, Photo, and Sketch. These datasets have large domain gaps. The classification results of the previous methods and our framework are shown in Table 1. Averagely, our framework consistently achieves the best performance in all three backbones

compared with previous works. Especially on Sketch, the accuracy of our framework is averagely $3\%$ better than the previous SOTA methods, showing superior performance. Our framework also obtains the best performance on Art painting in AlexNet and maintains the highest accuracy on Cartoon in ResNet50 (ours: $85.78\%$ vs. SOTA: $83.40\%$). This indicates that removing the domain-specific features from the input images is an effective approach for domain generalization. We can also study whether the domain-specific features would benefit or hurt the performance on the unseen target domain by comparing with mDSDI [8], as mDSDI uses the domain-specific features in addition to the domain-invariant features for domain generalization. We can see that our method significantly outperforms mDSDI on Cartoon and Sketch, and achieves a higher average classification performance than mDSDI in ResNet50. Meanwhile, mDSDI obtains better classification results than ours on Art and Photo. This shows that although Art and Photo may contain similar domain-specific features and these features would benefit each other, these domain-specific features would not benefit or even hurt Cartoon and Sketch.

VLCS also contains four domains. Table 2 shows the classification accuracy of the domain generalization methods using the AlexNet backbone. It can be seen that our framework obtains comparable performance to the best-performing methods, and outperforms the prior approaches on LabelMe and Sun. For Office-Home, ResNet18 is used as the backbone. The classification performance is shown in Table 3. Our framework outperforms the previous methods and achieves the best average performance. Besides, our framework obtains the best performance on Art. These experimental results demonstrate that removing the domain-specific features can significantly improve the generalization performance.

Table 2: Comparison with existing methods on VLCS using AlexNet backbone.

| Target | Baseline | CIDDG [30] | JiGen [9] | Epi-FCR [28] | MASF [13] | ER [45] | MetaVIB [14] | Stylized [7] | RSC [22] | LRDG (ours) |
|---|---|---|---|---|---|---|---|---|---|---|
| V | 66.27 | 64.38 | 70.62 | 67.10 | 69.14 | 73.24 | 70.28 | 68.18 | **73.93** | 68.95 |
| L | 61.81 | 63.06 | 60.90 | 64.30 | 64.90 | 58.26 | 62.66 | 60.77 | 61.86 | **65.53** |
| C | 96.17 | 88.83 | 96.93 | 94.10 | 94.78 | 96.92 | 97.37 | 96.86 | **97.61** | 96.85 |
| S | 63.78 | 62.10 | 64.30 | 65.90 | 67.64 | 69.10 | 67.85 | 63.42 | 68.32 | **69.27** |
| Avg. | 72.01 | 69.59 | 73.19 | 72.90 | 74.11 | 74.38 | 74.54 | 72.31 | **75.43** | 75.15 |

Table 3: Comparison with existing methods on Office-Home using ResNet18 backbones.

| Target | Baseline | D-SAMs [15] | JiGen [9] | DSON [38] | MixStyle [46] | SagNet [34] | Stylized [7] | RSC [22] | LRDG (ours) |
|---|---|---|---|---|---|---|---|---|---|
| A | 52.23 | 58.03 | 53.04 | 59.37 | 58.70 | 60.20 | 58.71 | 58.42 | **61.73** |
| C | 46.20 | 44.37 | 47.51 | 45.70 | **53.40** | 45.38 | 52.33 | 47.90 | 52.43 |
| P | 70.14 | 69.22 | 71.47 | 71.84 | **74.20** | 70.42 | 72.95 | 71.63 | 72.96 |
| R | 73.07 | 71.45 | 72.79 | 74.68 | **75.90** | 73.38 | 75.00 | 74.54 | 75.89 |
| Avg. | 60.41 | 60.77 | 61.20 | 62.90 | 65.50 | 62.34 | 64.75 | 63.12 | **65.75** |

## 3.3 Domain divergence

In this section, we investigate the distribution difference among the source domains and the target domain to demonstrate that our framework can effectively reduce domain divergence.

### 3.3.1 Source domain divergence

To investigate the distribution difference among the source domains, we compute the $\mathcal{H}$-divergence. Following the works of [6, 5], we can approximate the $\mathcal{H}$-divergence by a learning algorithm to discriminate between pairwise source domains. For example, with source domains $\mathcal{D}_s^i$ and $\mathcal{D}_s^j$, we label the samples of $\mathcal{D}_s^i$ by 1, and the samples of $\mathcal{D}_s^j$ by 0. We then train a classifier (e.g. linear SVM) to discriminate between these two domains. Given a test error $\varepsilon$ of this classifier, Proxy A-distance (PAD) is defined as $2(1 - 2\varepsilon)$, which can approximate the $\mathcal{H}$-divergence.

We follow the method from [19, 12, 1] to compute the PAD. For a pair of source domains, we combine these domains and construct a new dataset. This dataset is randomly split into two subsets of equal size. One subset is used for training and the other one is used for test. We train a collection of linear SVMs (with different values of regularization parameters) on the training set and compute the errors $\varepsilon$ of all the SVMs on the test dataset. The lowest error $\varepsilon$ is used to compute the PAD.

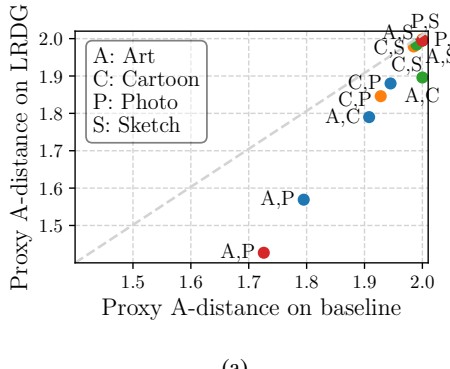 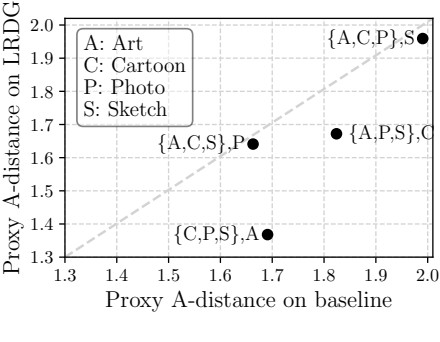

(a)                                      (b)

Figure 2: Proxy A-distance (PAD) on PACS. *x* axis: PAD computed upon the baseline model; *y* axis: PAD computed upon our framework. **(a)** PAD of pairwise source domains. *Blue dots*: PAD for Art, Cartoon, and Photo (Sketch as target domain). *Orange dots*: PAD for Cartoon, Photo, and Sketch (Art as target domain). *Red dots*: PAD for Art, Photo, and Sketch (Cartoon as target domain). *Green dots*: PAD for Art, Cartoon, and Sketch (Photo as target domain). **(b)** PAD of pairwise source-target domains. For a target domain (e.g. Art), the corresponding source domain is the closest mixture source domain (e.g. mixture of Cartoon, Photo, and Sketch) to the target domain. The PAD is computed on the mixture source domain and the target domain.

Fig. 2a compares the PAD of the raw source domains and the mapped source domains. The experiments are conducted on PACS with the AlexNet backbone. For the raw source domains, we extract features from the baseline model (i.e. the last pooling layer of AlexNet) to train the linear SVMs, while for the mapped source domains, we use features from our domain-invariant classifier. In the figure, each dot represents a pair of source domains (e.g. Art and Photo). It has two values: the PAD of the source domain pair obtained upon the baseline model (*x* axis) and the PAD of the same pair computed upon our framework (*y* axis). All the dots are below the diagonal meaning that the PAD values of the mapped pairwise source domains are lower than the raw pairwise source domains. With our framework, the mapped source domains become harder to be distinguished, indicating that removing the domain-specific features reduces the distribution difference among the source domains. This also proves that $\epsilon$ in the generalization risk bound (Eq. 11) would be reduced by our framework.

### 3.3.2 Source-Target domain divergence

We also investigate the distribution difference between the source domains and the target domain. Specifically, we measure the domain divergence between the target domain and the closest mixture source domain $\bar{\mathcal{D}}_t$ to the target domain. To obtain this mixture source domain, as defined in Sec. 2.3, we need to find $\pi_i$ for each source domain $D_s^i$, so that $\bar{\mathcal{D}}_t = \sum_{i=1}^{N} \pi_i \mathcal{D}_s^i$, where $0 \leq \pi_i \leq 1$ and $\sum_{i=1}^{N} \pi_i = 1$. Because $\pi_i$ can be any real value in the interval of $[0, 1]$, traversing all values to find the desired $\pi_i$ is impossible. Therefore, We limit the values of $\pi_i$ to the set of $\{0, 0.1, 0.2, \cdots, 0.9, 1\}$ (11 values in total), and find the setting of $\{\pi_i\}_{i=1}^{N}$ that can obtain $\bar{\mathcal{D}}_t$.

We traverse all possible settings of $\{\pi_i\}_{i=1}^{N}$ and obtain all possible mixture source domains $\bar{\mathcal{D}} = \sum_{i=1}^{N} \pi_i \mathcal{D}_s^i$. For each setting of $\{\pi_i\}_{i=1}^{N}$, we random sample $\pi_i n_t$ samples from each source domain $\mathcal{D}_s^i$ and concatenate all these samples into a mixture source dataset. $n_t$ is the number of samples in the target domain. By this design, each mixture source domain has an equal number of samples to the target domain. Similar to Sec. 3.3.1, we also train classifiers (i.e. linear SVMs) to discriminate between each mix-

Table 4: The settings of $\pi_i$ for the closest mixture source domains to the target domains on PACS.

| Source | | | Target |
|---|---|---|---|
| $D_s^1$: $\pi_1$ | $D_s^2$: $\pi_2$ | $D_s^3$: $\pi_3$ | $D_t$ |
| Cartoon: 0.2 | Photo: 0.8 | Sketch: 0 | Art |
| Art: 0.5 | Photo: 0.3 | Sketch: 0.2 | Cartoon |
| Art: 0.7 | Cartoon: 0.3 | Sketch: 0 | Photo |
| Art: 0.1 | Cartoon: 0.6 | Photo: 0.3 | Sketch |

ture source domain and the corresponding target domain. The linear SVMs are trained on image features extracted from the baseline model. We then use the test error to compute the PAD between

each mixture source dataset and the target dataset. The mixture source domain with the lowest PAD is the closest mixture source domain $\bar{\mathcal{D}}_t$ to the target domain. The detailed settings of $\pi_i$ for the closest mixture source domains to the corresponding target domains on PACS are listed in Table 4. For convenience, we denote the closest mixture source domain $\bar{\mathcal{D}}_t$ and the target domain $D_t$ together as a source-target domain pair.

Fig. 2b shows the PAD of the raw source-target domains and the mapped source-target domains. Similar to Sec. 3.3.1, for the raw source-target domains, we extract the image features from the baseline model to train the linear SVMs. To compute the PAD of the mapped source-target domains, we extracted the image features from our domain-invariant classifier to train the linear SVMs. In the figure, each dot represents a source-target domain pair (e.g. {Cartoon, Photo, Sketch}, Art). We can see that all the dots are below the diagonal. The PAD values of the mapped source-target pairs are lower than the raw source-target pairs. This indicates that the distribution difference between the source domains and the target domain is reduced by our framework. $\gamma$ in the generalization risk bound (Eq. 11) would be lowered. Removing the domain-specific features from the source domains can also reduce the distribution difference between the source domains and the target domain.

In summary, our framework can reduce the distribution difference not only among the source domains but also between the source domains and the target domain. This also demonstrates that our framework could effectively lower the value of the generalization risk bound by reducing $\epsilon$ and $\gamma$.

## 4 Related work

Domain generalization is a challenging task that requires models to be well performed on unseen domains. One common approach is to learn domain-invariant features among the source domains. Previous methods aim to distill the domain-invariant features, but they do not clearly inform the DNNs that the domain-specific features shall be effectively removed. Muandet et al. [33] propose to reduce the domain dissimilarity by a kernel-based method. Ghifary et al. [18] reduce dataset bias by extracting features that are shared among the source domains with a multi-task autoencoder network. Li et al. [29] utilize Maximum Mean Discrepancy (MMD) on adversarial autoencoders to align the distributions across source domains. Li et al. [30] design an end-to-end conditional invariant deep neural network that minimizes the discrepancy of conditional distributions across domains. Arjovsky et al. [3] develop Invariant Risk Minimization (IRM) that uses a causal mechanism to obtain the optimal invariant classifier upon the representation space. Chattopadhyay et al. [11] propose to learn domain-specific binary masks to balance the domain-invariant and domain-specific features for the prediction of unseen target domains. Zhao et al. [45] propose an entropy regularization method to learn the domain-invariant conditional distributions by using a classification loss and a domain adversarial loss. Du et al. [14] develop a probabilistic meta-learning method that learns domain-invariant representations with meta variational information bottleneck principle derived from variational bounds of mutual information. Mahajan et al. [31] assume that domains are generated by mixing causal and non-causal features and that the same object from different domains should have similar representations. Based on this, they propose a new method called MatchDG to build a domain-invariant classifier by matching similar inputs. Rame et al. [35] match the gradients among the source domains to minimize domain invariance. Unlike the above works, Bui et al. [8] assume that, besides the domain-invariant features, some domain-specific features also provide useful information for the target domain. However, this cannot always be guaranteed since the target domain is unseen. For example, the backgrounds in the domain Photo may benefit the domain Art, but they would not benefit or even hurt the domain Sketch. Our framework follows the common assumption that the domain-invariant features are generalized across domains, regardless of the effect of the domain-specific features [30, 3, 45].

Recent papers demonstrate that CNNs tend to classify objects based on features from superficial local textures and backgrounds, while humans rely on global object shapes for classification [23, 17]. To address this issue, some methods aim to capture the global object shapes from the images. These methods are proposed based on the assumption that the local textures and backgrounds are the domain-specific features, and the global object shapes are the domain-invariant features. Wang et al. [42] extract semantic representations by penalizing features extracted with gray-level co-occurrence matrix (GLCM) which are sensitive to texture. Wang et al. [41] penalize the earlier layers of CNNs from learning local representations and make the CNNs rely on the global representations for classification. Although addressing the superficial local features is a promising approach, the superficial local

features may be one kind of domain-specific features and other forms of domain-specific features may also exist. Compared with these methods, our framework is proposed to address the more general domain-specific features rather than the superficial local features.

## 5 Conclusion

In this work, we propose a new approach that aims to explicitly remove domain-specific features for domain generalization. To this end, we develop a novel domain generalization framework that learns the domain-invariant features by actively removing the domain-specific features from the input images. We also experimentally verify the reduced domain divergence among the source domains and the target domain brought by our approach. Experiments show that our framework achieves strong performance on various datasets compared with existing domain generalization methods.

Despite the advantages of our framework, it has some potential limitations to be further addressed. We need to train the same number of domain-specific classifiers as the source domains. When there are more source domains, more computational resources will be required to train the domain-specific classifiers. This may be addressed by designing a novel domain-specific classifier that can learn the domain-specific features of multiple source domains simultaneously. Another limitation of our framework is that it cannot remove the domain-specific features of the unseen target domain. These domain-specific features should also be removed since they would negatively affect the classification performance. For example, our framework performs slightly worse than the baseline when Photo is the target domain (as shown in Table 1). This may be because Photo contains rich domain-specific features compared with the source domains, and our framework would make incorrect predictions due to these domain-specific features. Besides, this result also shows that domain-specific knowledge is useful for Photo. As the target domain is not available during training, how to remove the domain-specific features from the target domain and whether to retain the domain-specific features of the source domains will be challenging issues to be addressed. One possible future work may be to remove the domain-specific features in a latent feature space. To achieve this, the framework may need to be adjusted, including the domain-specific classifiers and the domain-invariant classifier. The encoder-decoder network incurs extra computational overhead, but performing on a latent space may have the benefit that we may no longer need the encoder-decoder network and the overall framework can be computationally more efficient.

## Acknowledgment

Yu Ding was supported by CSIRO Data61 PhD Scholarship and the University of Wollongong International Postgraduate Tuition Award. This research was undertaken with the assistance of resources and services from the National Computational Infrastructure (NCI) and the CSIRO Accelerator Cluster-Bracewell.

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
