# Domain Generalization by Learning and Removing Domain-specific Features – Appendix

**Yu Ding**
University of Wollongong
yd624@uowmail.edu.au

**Lei Wang**
University of Wollongong
leiw@uow.edu.au

**Bin Liang**
University of Technology Sydney
Bin.Liang@uts.edu.au

**Shuming Liang**
University of Technology Sydney
Shuming.Liang@uts.edu.au

**Yang Wang**
University of Technology Sydney
Yang.Wang@uts.edu.au

**Fang Chen**
University of Technology Sydney
Fang.Chen@uts.edu.au

## A  Implementation details

### A.1  Data processing

The training data are processed with a fixed pipeline. The input images are resized and cropped to $224 \times 224$, then randomly transformed by horizontal flip, color jitter, and grayscale, and finally normalized with the standard ImageNet [9] channel statistics. This setting is widely used in previous works in domain generalization [1, 3, 4].

### A.2  Training setting

The source data are split into a training set and a validation set according to [1, 3, 4]. The Stochastic Gradient Descent is used as the optimizer with weight decay $10^{-4}$ and momentum 0.9. The learning rate is chosen from $\{10^{-3}, 10^{-4}\}$ according to the performance of the validation set. For the main results in our experiments, we use three seeds $\{8, 9, 10\}$ to run the experiments and the final results are obtained by averaging these three runs. All the models are trained on NVIDIA V100 and P100.

### A.3  Hyperparameters

Our framework has three hyperparameters including $\lambda_1$, $\lambda_2$ and $\lambda_3$. We set $\lambda_1 = 1$ for all the experiments. For $\lambda_2$ and $\lambda_3$, we follow the literature [3, 1, 2] and directly use the leave-one-domain-out cross-validation to select their values. We use grid search to find the values for $\lambda_2$ and $\lambda_3$. $\lambda_2$ is selected from $\{0.1, 0.01\}$ and $\lambda_3$ is selected from $\{1, 0.1, 0.01, 0.001, 0.0001\}$.

## B  License of existing assets

The existing datasets and codes used in this paper are publicly available. The licenses are listed as follows.

Datasets: Office-Home [14] is for non-profit academic research and education only. We cannot find the license for PACS [6] and VLCS [13].

36th Conference on Neural Information Processing Systems (NeurIPS 2022).

Codes: AlexNet, ResNet18, and ResNet50 are pretrained with ImageNet in torchvision. They are under BSD 3-Clause License. U-net is under GNU General Public License v3.0.

## C Domain-specific and domain-invariant features

To demonstrate the ability of our framework in capturing the domain-invariant features, we first compare our domain-specific classifier with the one proposed by Epi-FCR [7], and then visually illustrate the domain-specific and domain-invariant features learned by our framework.

### C.1 Comparison of domain-specific classifiers

Epi-FCR [7] also introduces a domain-specific classifier that is trained by the classification loss of a source domain. However, this method cannot ensure that its domain-specific classifier would not use the domain-invariant features for classification. Unlike Epi-FCR, in addition to minimizing the classification loss for each source domain, our domain-specific classifier also maximizes the classification uncertainty on the remaining source domains. Our domain-specific classifier is therefore designed to only learn the domain-specific features. In Table 1, we show the performance of using the domain-specific classifier from Epi-FCR and our framework on PACS. The performance of Epi-FCR is obtained by training our encoder-decoder network and domain-invariant classifier with the domain-specific classifiers from Epi-FCR. We can see that the prediction performance using our domain-specific classifier consistently outperforms that obtained by Epi-FCR, which shows that our domain-specific classifier can better learn the domain-specific features than the one from Epi-FCR.

Table 1: Prediction accuracy (%) on PACS with different domain-specific classifiers.

| Method | A | C | P | S | Avg. |
|---|---|---|---|---|---|
| Epi-FCR [7] | 64.32 | 71.97 | 88.03 | 71.38 | 73.92 |
| LRDG (ours) | **72.01** | **73.12** | **89.50** | **74.86** | **77.37** |

### C.2 Visualization of domain-specific and domain-invariant features

To intuitively illustrate the domain-specific features and the domain-invariant features, we show the Grad-CAM and Guided Grad-CAM [10] visualization of the images from House and Person categories in Fig. 1 and Fig. 2. Grad-CAM is a visualization technique that locates the important regions in the image for prediction. Guided Grad-CAM combines Grad-CAM with Guided backpropagation [12] to obtain a high-resolution gradient visualization. For this experiment, the source domains are Photo, Art painting, and Cartoon, and the target domain is Sketch.

In Fig. 1, we show the Grad-CAM and Guided Grad-CAM visualization of the images from the source domains obtained from four models including our domain-invariant classifier, the baseline classifier, our domain-specific classifier, and the domain-specific classifier from EPI-FCR. The baseline classifier is the baseline model trained by minimizing the cross entropy loss on all source domains. We compare these classifiers to show that they recognize different features for inference. Our domain-invariant classifier focuses on the features of triangular roofs or the top of the windows to recognize houses and locates the features from hairlines or head shapes to recognize a person. These features exist in all source domains and can be treated as domain-invariant features. The baseline classifier focuses on the doors, windows, and backgrounds of the houses and the whole face of the person. It captures both the domain-specific and the domain-invariant features. For example, backgrounds (e.g. grassland, trees, and flowers) and face details do not always exist in all source domains while head shapes belong to all source domains. The domain-specific classifier from Epi-FCR uses features that are similar to the baseline classifier and tends to use domain-invariant features for classification. Our domain-specific classifier uses features (e.g. grassland, trees, and flowers for House, and the lower faces for Person) that belong to the specific domains. It can better capture the domain-specific features than that from Epi-FCR.

Fig. 2 shows the Grad-CAM and Guided Grad-CAM visualization of the images from the target domain Sketch. We compare our domain-invariant classifier and the baseline classifier. The prediction accuracy of each classifier is also shown. Our domain-invariant classifier can capture triangular

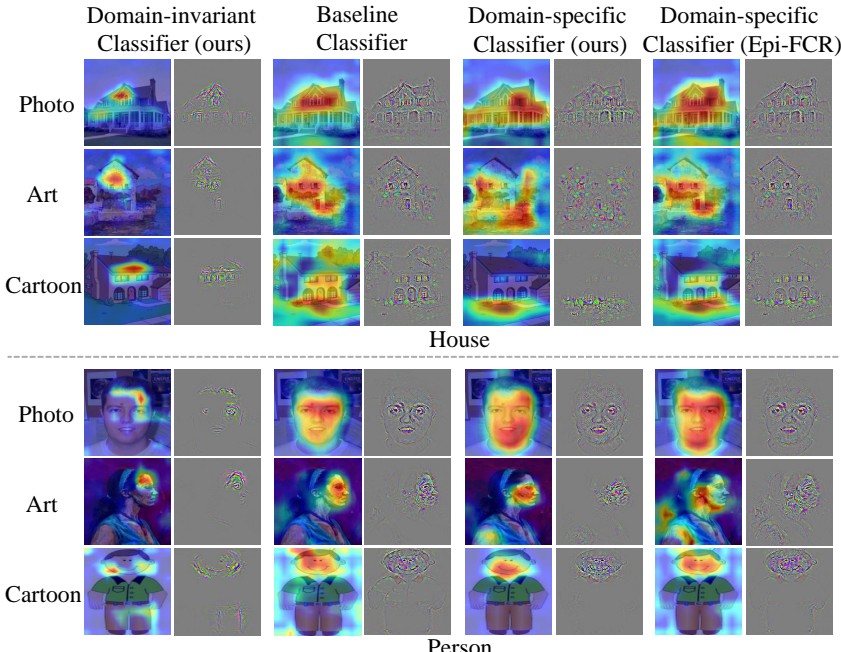

Figure 1: Grad-CAM and Guided Grad-CAM for House and Person on source domains with different methods. For each category (House or Person), three images from Photo (1st row), Art (2nd row), and Cartoon (3rd row) are shown with different methods. For each method, the left and right images are the visualization results of Grad-CAM and Guided Grad-CAM, respectively. *Domain-invariant Classifier (Ours)* is the domain-invariant classifier proposed in this paper. *Baseline Classifier* is the classifier obtained from the baseline method. *Domain-specific Classifier (Ours)* is the domain-specific classifier proposed in this paper. *Domain-specific Classifier (Epi-FCR)* is the domain-specific classifier used by Epi-FCR.

roofs to recognize houses and head shapes to recognize persons, but the baseline classifier hardly extracts useful features to correctly identify objects. As illustrated in the figure, the baseline classifier categorizes the houses as Person or Elephant and classifies the persons as Dog, Elephant, or Giraffe. With our domain-invariant classifier, the classification accuracy for House is increased by about $46\%$ and the accuracy for Person is improved by almost $54\%$. This demonstrates the advantage of our framework for learning domain-invariant features compared with the baseline classifier.

## D   Loss functions

In this experiment, we compare the performance of different loss functions including the uncertainty loss $L_U$ and the reconstruction loss $L_R$. For $L_U$, we evaluate two losses including the entropy loss that measures the entropy of the posterior probability of classification and the least likely loss [8] that aims to predict the least likely class. For $L_R$, we evaluate three losses including the $l_1$ loss, $l_2$ loss and perceptual loss [5]. The $l_1$ or $l_2$ loss measures the pixel-wise similarity while the perceptual loss measures the semantic similarity between images. Johnson et al. [5] proposed two perceptual loss functions: feature reconstruction loss and style reconstruction loss. We only use the feature reconstruction loss because we aim to reconstruct the semantic features instead of the style of the images. To compute the feature reconstruction loss, we use the VGG [11] pre-trained by ImageNet [9] as the loss network [5]. Since the domain of ImageNet is different from the source domains, we first fine-tune the loss network with the source domains and further use it to compute the feature reconstruction loss.

In Table 2, we show the prediction accuracy of these loss functions on PACS with the AlexNet backbone. As shown in the table, the entropy loss consistently achieves better performance than the least likely loss. The performance of the $l_1$ and $l_2$ reconstruction loss is comparable, but the

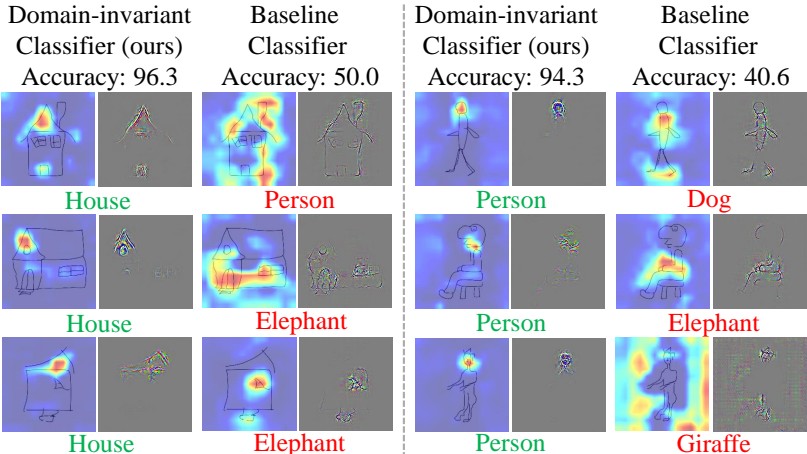

| Domain-invariant Classifier (ours) Accuracy: 96.3 | Baseline Classifier Accuracy: 50.0 | Domain-invariant Classifier (ours) Accuracy: 94.3 | Baseline Classifier Accuracy: 40.6 |

House — Person — Person — Dog

House — Elephant — Person — Elephant

House — Elephant — Person — Giraffe

Figure 2: Grad-CAM and Guided Grad-CAM for House and Person on the target domain (Sketch) with different methods. For each category (House or Person), three images from Sketch are shown with different methods. For each method, the left and right images are the visualization results of Grad-CAM and Guided Grad-CAM, respectively. *Domain-invariant Classifier (Ours)* is the domain-invariant classifier proposed in this paper. *Baseline Classifier* is the classifier obtained from the baseline method. The classes of the images predicted by each classifier are below each image. The prediction accuracy (%) is also shown.

Table 2: Prediction accuracy (%) on PACS with loss functions $L_U$ and $L_R$. *EL*: entropy loss; *LLL*: least likely loss; $l_2$: $l_2$ reconstruction loss; $l_1$: $l_1$ reconstruction loss; *PL*: perceptual loss with the loss network pre-trained by ImageNet; *PL (src)*: perceptual loss with the loss network fine-tuned by the source domains.

| $L_U, L_R$ | A | C | P | S | Avg. |
|---|---|---|---|---|---|
| EL, $l_2$ | **72.01** | 73.12 | **89.50** | **74.86** | **77.37** |
| EL, $l_1$ | 67.44 | **74.11** | 88.93 | 74.65 | 76.28 |
| EL, PL | 70.12 | 71.43 | 88.01 | 73.49 | 75.76 |
| EL, PL (src) | 68.36 | 72.13 | 88.33 | 74.58 | 75.85 |
| LLL, $l_2$ | 68.37 | 71.24 | 87.76 | 73.31 | 75.17 |

latter has better average accuracy. The performance of the perceptual loss is worse than that of the $l_2$ reconstruction loss. Even though the loss network is fine-tuned on the source domains, the performance of the perceptual loss shows no improvement. Overall, we use the entropy loss and the $l_2$ reconstruction loss as the default uncertainty loss and reconstruction loss function, respectively.

# E   Visualization of mapped images

Figure 3 shows the mapped images **z** generated from the encoder-decoder network with the proposed framework trained with various settings of $(\lambda_2, \lambda_3)$. The input image is an image of a house from the Art painting dataset. In this experiment, the source domains are Art painting, Cartoon, and Photo, and the target domain is Sketch. The hyperparameters $\lambda_2$ and $\lambda_3$ determine the level of removing the domain-specific features learned from the domain-specific classifiers. The setting of larger $\lambda_2$ and smaller $\lambda_3$ heavily removes the domain-specific features while the setting of smaller $\lambda_2$ and larger $\lambda_3$ retains the raw input images. As seen from the figure, when $\lambda_2 = 0.01$ and $\lambda_3 = 1$, the mapped images are almost similar to the input image. When $\lambda_2 = 20$ and $\lambda_3 = 0.0001$, only edges of the raw image tend to remain in the mapped images, and the color features become relatively insignificant.

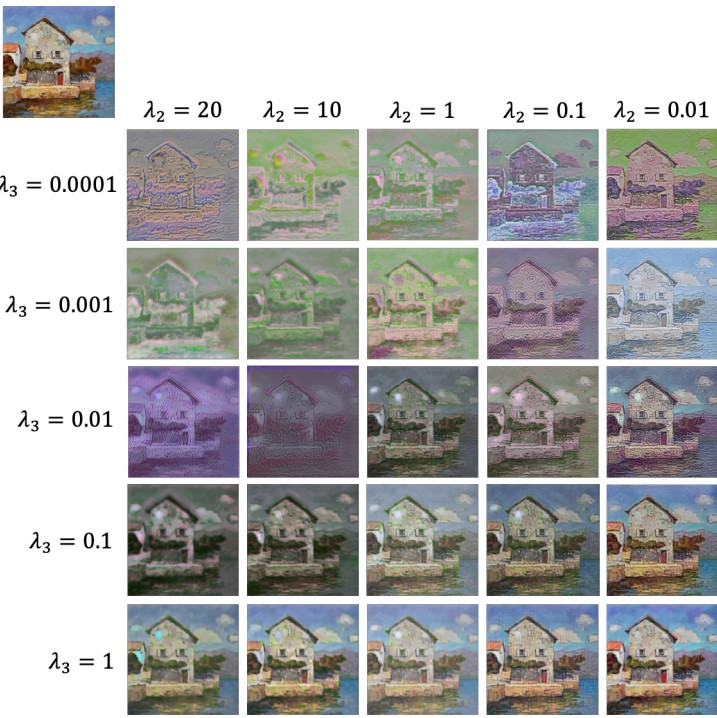

Figure 3: Mapped images with various settings of $(\lambda_2, \lambda_3)$. The image on the top is the raw image. The remaining images are the mapped images from the encoder-decoder network with the proposed framework trained with the corresponding settings of $(\lambda_2, \lambda_3)$. The rows show the values of $\lambda_3$ and the columns show the values of $\lambda_2$.