# OpenReview forum: "Domain Generalization by Learning and Removing Domain-specific Features"
_NeurIPS.cc/2022/Conference — NeurIPS 2022 Accept_

### Official Review · Reviewer_8K1L · 2022-07-05

**Rating:** 8
**Confidence:** 3
**Soundness:** 3 good
**Presentation:** 4 excellent
**Contribution:** 4 excellent

**Summary:**

The manuscript describes a new framework for training models that are general for multiple domains (domain generalisation). Their protocol is to (1) train one classifier for each domain, which are made domain-specific by enforcing a random guess for domains that they are not specialised to, (2) train an auto encoder where the decoding process is constrained to generate images that “confuse” all of the domain specific classifiers and at the same time (3) train a new classifier on top of those domain-agnostic images.

**Questions:**

Why does Figure 2 not include PAD of pairwise domains with Photo and Cartoon as target domains?

Did the authors consider performing a similar method on a latent space instead of directly on the image space? (This question or its answer do not interfere with my rating)


**Limitations:**

The limitations of the model are addressed in the Conclusion. The authors discuss the need for training separate classifiers for each source domain and that the model is not able to remove domain-specific features that regard unseen domains. I would add the limitation that some domain-specific knowledge is useful for some domains, as we can see that this model does not outperform others on more complex domains such as Photo and sometimes Art.

EDIT (post-discussion): I will also add this here for completeness. Per the discussion below and with other reviewers, the authors have acknowledged that the removal of domain-specific features is conditional on these features being learned by the domain-specific classifiers via the optimisation in Eq. (8). This is expected to work given the design of Eq. (8) but it is not *guaranteed*, and therefore this assumption configures an expected limitation of the work that future readers and users should keep in mind.

**Strengths And Weaknesses:**

The paper is well written and easy to follow. The presented idea for generating domain-invariant images and training domain specific classifiers is intuitive and the entire package seems to bring about improvements on average to performance on the baselines datasets. The authors have made comparisons against state-of-the-art studies and compared across different backbones.

The use the generalisation bound, and of the Proxy A-distance (PAD) is very welcome here for showing that the model truly generalises. However, the application used by the authors employ the features of an AlexNet network for the unmapped domains and the features of their domain-invariant classifier for the mapped domains; this makes it hard to assess whether their mapping was effective in reducing the H-divergence between domains. It would probably be more adequate to use the same features for both to allow for a fairer comparison. I understand this is not a simple choice given the features of an AlexNet will be more discriminative for the photo domain (and this will probably also apply for other pre-trained networks) and the features of the domain-invariant classifier would be less discriminative for unmapped domains since it was trained for mapped versions.

Even though it is the intuitive conclusion that the framework would remove domain-specific features, it is not guaranteed to do so. Different classifiers could work with different manifolds of the original image space to do their classification and the uncertainty loss could force each classifier to use their own “space” to perform classification, ignoring or not domain-specific features from other domains; this is similar to how SVMs with different kernels could solve the same classification tasks and even use spaces with the same dimensionality. This non-guarantee is important to state since the authors state that they reduce \lambda (lines 174-176) and “prove” that they reduce e in the generalisation risk bound; maybe those statements could be toned down with “probably” or “indicates that … reduces”.

---

> ### Author Response · Authors · 2022-08-02
> **Response to Reviewer 8K1L**
>
> We thank the reviewer for the detailed suggestions. We have revised our paper based on the comments.
>
> **1. Using the same features for the mapped and unmapped images to compute Proxy A-distance (PAD).**
>
> We thank the reviewer for pointing this out. However, the domain-invariant classifier and the encoder-decoder network are jointed trained, and therefore they shall be viewed as a whole. All the images need to pass the encoder-decoder network and then be fed into the domain-invariant classifier. Directly feeding the original images to the domain-invariant classifier to extract features will be inappropriate. Therefore, in the experimental study, we have to use the features extracted from the baseline for the unmapped images and use the ones extracted from the domain-invariant classifier for the mapped images.
>
> We appreciate that the reviewer has also been aware of this situation.
>
> **2. The domain-specific features are not guaranteed to be removed.**
>
> Thanks for the comments and we have revised the paper to adjust the tone accordingly. In addition, we would like to mention that, as in our answer to the 1st question of Review T9VV, our framework is able to remove the domain-specific features as long as they are learned by the domain-specific classifiers although we cannot guarantee to completely remove all the domain-specific features.
>
> **3. Figure 2 does not include PAD of pairwise domains with Photo and Cartoon as target domains.**
>
> In the paper, we did not include the PAD for Photo and Cartoon because their results show they follow the same trend as Art and Sketch, i.e., the PAD of the mapped pairwise source domains are lower than the raw pairwise source domains. We add the PAD of Photo and Cartoon in Figure 2 in the revised paper. The results still support the analysis of the domain divergence.
>
> **4. A similar method performed on a latent space.**
>
> Thanks for your suggestion. We can also remove the domain-specific features in a latent feature space. To achieve this, the framework may need to be adjusted, including the domain-specific classifiers and the domain-invariant classifier. Also, performing on a latent space may have the benefit that we may no longer need the encoder-decoder network and the overall framework can be computationally more efficient. We have put the discussion of this alternative way in the Conclusion section 5.

---

> > ### Comment · Reviewer_8K1L · 2022-08-09
> > **Thank you for the response!**
> >
> > I thank the authors for their detailed responses to both my concerns and other reviewers' as well. I believe this specific assumption should be made explicitly on the manuscript: "our framework is able to remove the domain-specific features as long as they are learned by the domain-specific classifiers". This would help a reader/user to have the right expectations when training a similar model; e.g. if it doesn't work, a good place to start would be checking this assumption by looking at each domain-specific classifier's metrics.
> >
> > Assuming this will be incorporated, I will raise my rating to strong accept. I hope this gets accepted. Good work authors, thank you for the discussion and trusting us reviewers with your paper.

---

> > > ### Author Response · Authors · 2022-08-09
> > > **Thanks for your feedback!**
> > >
> > > The suggested assumption makes our framework and the whole paper more rigorous. We will surely incorporate it into our paper. Thanks again for all the comments.

---

### Official Review · Reviewer_MBUN · 2022-07-11

**Rating:** 6
**Confidence:** 4
**Soundness:** 3 good
**Presentation:** 3 good
**Contribution:** 2 fair

**Summary:**

This paper introduces a novel framework, namely learning and removing domain-specific features for generalization (LRDG), allowing the trained model can extract only domain-invariant features in order to improve the out-of-domain generalization performance. In particular, in LRDG, for each source domain, the training process of the domain-specific classifier is based on the classification loss and the uncertainty loss on other source domains. The experimental results show that the proposed method can produce better classification accuracy compared to several existing methods.

**Questions:**

1. A mathematical definition for domain-specific features?
2. A theoretical guarantee capability to learn the real domain-specificity of the proposed LRDG?
3. A comparison with the method in [3], which indicates the usefulness of domain-specific features in domain generalization?

**Strengths And Weaknesses:**

Originality: The proposed framework for learning domain-specific features based on an autoencoder and uncertainty loss in the paper is novel in domain generalization.

Quality: The technical contributions of the paper are relatively insignificant due to the lack of a mathematical definition for domain-specific features and a theoretical guarantee for the capability to learn the real domain-specificity of the proposed LRDG. Although I appreciate the theoretical result about the generalization bound in Theorem 2, it does not support the main claim of the paper.

Clarity: The paper is quite well-written and easy to follow.

Significance: My major concern is about the real benefit of the proposed method. In particular, compared to the direct domain-invariant learning approaches (see e.g., [1,2]) the LRDG framework seems to be more complicated and more computationally expensive since requires the training of an autoencoder.  Moreover, the main idea of LRDG is to learn and eliminate the domain-specific features, while some recently published papers (e.g., [3]) show that effectively combining domain-specific and domain-invariant features are also useful for domain generalization.

[1] @inproceedings{li2018deep, title={Deep domain generalization via conditional invariant adversarial networks}, author={Li, Ya and Tian, Xinmei and Gong, Mingming and Liu, Yajing and Liu, Tongliang and Zhang, Kun and Tao, Dacheng}, booktitle={Proceedings of the European Conference on Computer Vision (ECCV)}, pages={624--639}, year={2018} }

[2] @inproceedings{hu2020domain, title={Domain generalization via multidomain discriminant analysis}, author={Hu, Shoubo and Zhang, Kun and Chen, Zhitang and Chan, Laiwan}, booktitle={Uncertainty in Artificial Intelligence}, pages={292--302}, year={2020}, organization={PMLR} }

[3] @article{bui2021exploiting,
  title={Exploiting domain-specific features to enhance domain generalization},
  author={Bui, Manh-Ha and Tran, Toan and Tran, Anh and Phung, Dinh},
  journal={Advances in Neural Information Processing Systems},
  volume={34},
  pages={21189--21201},
  year={2021}
}

---

> ### Author Response · Authors · 2022-08-02
> **Response to Reviewer MBUN**
>
> We thank the reviewer for the detailed suggestions. We have revised our paper based on the comments.
>
> **1. A mathematical definition for domain-specific features.**
>
> As mentioned in our answer to the 1st question of Review T9VV, our framework is able to remove the domain-specific features as long as they are learned by the domain-specific classifiers via the optimization in Eq. (8). Therefore, the domain-specific features in our work can be mathematically defined as follows.
>
> To facilitate the definition, we explicitly express a domain-specific classifier $F_i$ as a feature extractor $G_i$ with the parameter $\theta_i^G$ and a label predictor $H_i$ with the parameter $\theta_i^H$. That is, $F_i=H_i \circ G_i$ and $\theta_i=\{\theta_i^G,\theta_i^H\}$. By this definition, $F_i(x_j^i; \theta_i)$ in Eq.(1) (classification loss) is replaced with
> $H_i(G_i(x_j^i; \theta_i^G); \theta_i^H)$ ,
> and $F_i(x_j^k; \theta_i)$ in Eq.(2) (uncertainty loss) is replaced with
> $H_i(G_i(x_j^k; \theta_i^G); \theta_i^H)$.
>
> The features obtained from $G_i$, i.e. $G_i(x_j^i; \theta_i^G)$,  are the domain-specific features. Mathematically, it can be expressed as ${\theta_i^G}^{*} = \arg\min (L_{C}^{F_i} + \lambda_1 L_{U}^{F_i})$, where $L_{C}^{F_i}$ is the classification loss (Eq.(1)) and $L_{U}^{F_i}$ is the uncertainty loss (Eq.(2)).
>
> **2. A theoretical guaranteed capability to learn the real domain-specificity.**
>
> Using the definitions of domain-specific features in the last answer, we now assume that the domain-specific classifier $F_i$ learns domain-invariant features to classify images from domain $i$, i.e. $G_i$ extracts domain-invariant features $G_i(\mathbf{x}_j^i; \theta_i^G)$ for classification.
>
> By the meaning of domain-invariant features, we know that the remaining domains shall also share the domain-invariant features with domain $i$. In other words, $G_i$ will also extract the domain-invariant features $G_i(x_j^k; \theta_i^G)$ for any domain $k$ ($ k \neq i $). These domain-invariant features $G_i(x_j^k; \theta_i^G)$ will increase the classification certainty of $F_i$ for domain $k$.
>
> This clearly contradicts the objective function of maximizing the uncertainty loss defined in Eq. (2). Therefore, maximizing the uncertainty loss prevents the domain-specific classifier from using domain-invariant features and forces it to learn the domain-specific features for classification.
>
> Certainly, rigorous theoretical proof will be sought in our future work.
>
> **3. Complicated and computationally expensive due to the training of an encoder-decoder network.**
>
> Our framework trains an encoder-decoder network, which seems to be more complicated than [R1, R2], as pointed out in the review comment. Meanwhile, we would like to mention that in the literature on domain generalization, it is not uncommon for a proposed approach (say, [18,28,29] (references from our paper)) to train multiple neural networks (in addition to the backbone) so as to improve classification performance. For example, the method in [29] needs to train an autoencoder to learn domain-invariant representations. Reducing this kind of computational complexity is an important issue to be addressed in this research area. However, it is currently not the focus of our paper, and we will pursue it in future work.
>
> **4. A comparison with the method in [R3].**
>
> The work of [R3] assumes that the domain-specific features provide useful information for the target domain. However, this cannot always be guaranteed. For example, the backgrounds in the domain Photo may benefit the domain Art, but they would not benefit or even hurt the domain Sketch (as its images only contain black lines).
>
> Differently, our work only assumes that the domain-invariant features are generalized across domains (as is also taken in the literature [R1,R2]), regardless of the effect of the domain-specific features. In other words, the work of [R3] and our work are based on two different assumptions.
>
> Following the suggestion, we compare the prediction accuracy with mDSDI [R3] on PACS. As shown in the table below, our method significantly outperforms mDSDI on Cartoon and Sketch and produces a higher average classification performance on the four target domains.
> Meanwhile, mDSDI achieves better results than ours on Art and Photo. This can be explained as follows: Art and Photo may contain similar domain-specific features and these features would benefit each other. Note that this is not the case for Cartoon and Sketch.
>
> |Method | Artpaint | Cartoon | Photo | Sketch | Avg. |
> | -- | -- | -- | -- | -- | -- |
> |mDSDI [C] | **87.7** | 80.4 | **98.1** | 78.4 | 86.2 |
> |ours | 86.5 | **85.7** | 95.5 | **86.5** | **88.6** |
>
> [R1] - Deep domain generalization via conditional invariant adversarial networks. ECCV 2018
>
> [R2] - Domain generalization via multidomain discriminant analysis. Uncertainty in Artificial Intelligence 2020
>
> [R3] - Exploiting domain-specific features to enhance domain generalization. NeurIPS 2021

---

> > ### Comment · Reviewer_MBUN · 2022-08-06
> > **Upgrading the score to 6**
> >
> > Thank you for your effort to address my concerns, especially for providing an empirical comparison with mDSDI [3] - I hope that those new results will be added to the revised version of the paper together with a clear clarification about the main difference between your paper and [3].  Regarding my concern about a theoretical guarantee of the capability to learn the real domain-specificity, I think that is still an important paper and should be included in the paper due to the standard quality of a NeurIPS paper.
> >
> > I decide to increase the score of the paper to 6.

---

> > > ### Author Response · Authors · 2022-08-07
> > > **Thanks for your feedback**
> > >
> > > We would like to thank the reviewer for the feedback. We will surely add the experimental comparison with mDSDI [R3] and clarify our differences from it in the revised version of our paper. Your valuable suggestions are sincerely appreciated.

---

### Official Review · Reviewer_T9VV · 2022-07-11

**Rating:** 4
**Confidence:** 3
**Soundness:** 2 fair
**Presentation:** 3 good
**Contribution:** 2 fair

**Summary:**

The paper proposes a domain generalization method. The key idea is to remove domain-specific features by first training domain-specific classifiers for all domains then training encoder-decoder network that transforms an input image to a domain-invariant version based on them. Specifically, the encoder-decoder network is trained together with another domain-invariant classifier so that the domain-specific classifiers cannot discriminate the classes of transformed images but the domain-invariant classifier can classify them. Experiments show that the model improves the generalization performance from the baselines on PACS, VLCS, and Office-Home with several different backbone architectures.

**Questions:**

- It would be interesting to visualize output of the encoder-decoder network. It will provide more insights how the method behaves.
- How does the encoder-decoder architecture affect the performance?
- Which data augmentation was used for training? Was the same augmentation used for all the comparing methods?

**Limitations:**

The authors did not address the limitations and potential negative societal impact of their work.

**Strengths And Weaknesses:**

1. Strengths

- The idea to learn domain-invariant features has been studied in several existing work. Among them, the paper looks similar to the Epi-FCR [28], which also uses domain-specific classifiers to train a domain-invariant feature extractor. This paper explicitly designs a training scheme to remove domain-specific features and the effect looks pretty clear.
- The proposed method consistently improves the baseline on PACS, VLCS, and Office-Home with different backbone architectures.
- The paper is clearly written and easy to follow.

2. Weaknesses
- I think maximizing the classifier uncertainty of a domain-specific classifier does not necessarily guarantee that the domain-specific features are removed. For example, assume an input image has two channels and the trained domain-specific classifier F_1 uses only the first channel. Since the classifier is fixed, the encoder-decoder network can cheat to include domain-specific features in the second channel, so that it can perform well on the classifier F, which was originally intended to be domain-invariant. At the same time, encoder-decoder network can be trained to output the first channel constant so that the classifier F_1 cannot discriminate classes properly.
- Minor weakness is that the method needs to pass the input image through the encoder-decoder network and it makes an overhead in inference time.

---

> ### Author Response · Authors · 2022-08-02
> **Response to Reviewer T9VV**
>
> We thank the reviewer for the detailed suggestions. We have revised our paper based on the comments.
>
> **1. Maximizing the classifier uncertainty of a domain-specific classifier does not necessarily guarantee that the domain-specific features are removed.**
>
> Thanks for the comment. The assumption made in the review comment is that there may exist two channels of domain-specific features in an image and that our domain-specific classifier only learns the domain-specific features in the first channel.
>
> If it is indeed this case, then our framework will be able to remove the domain-specific features from the first channel, while the mapped images could still contain the domain-specific features from the second channel as well as the domain-invariant features. That is to say, our framework is able to remove the domain-specific features as long as they are learned by the domain-specific classifiers via the optimization in Eq. (8), although we cannot guarantee to completely remove all the domain-specific features, especially those that are not learned by any domain-specific classifiers.
>
> In addition, by our proactive removal of the domain-specific features (even partially), the domain-invariant features could become more pronounced in the mapped images than in the original images. This would in turn make our model better prepared for the domain-invariant classifier to learn the domain-invariant features for classification.
>
> **2. Overhead in inference time by the encoder-decoder network.**
>
> We agree that introducing an encoder-decoder network (which is just the commonly used U-Net) into our approach indeed incurs extra computational overhead. However, this overhead can be well justified by the improved classification performance achieved by our approach, as experimentally demonstrated. To address this computational issue, we could 1) explore other more efficient networks for our approach or 2) directly remove domain-specific features in a latent feature space instead of explicitly using the mapped images, as mentioned by Reviewer 8K1L. We have highlighted this as future work in the Conclusion section of the revised paper.
>
> **3. Visualization of the output of the encoder-decoder network.**
>
> Following the suggestion, we add the visualization of the output images from the encoder-decoder network in the supplementary materials. As shown in Figure 3 in Section E, the output images under various settings of $\lambda_2$ and $\lambda_3$ are displayed. From this figure, we can see that our approach can effectively remove the domain-specific features at various levels.
>
> **4. How the encoder-decoder architecture affects the performance?**
>
> In principle, our approach can work with any capable encoder-decoder network. In this paper, we follow the literature (say, [32] in our paper) to use the common U-net to demonstrate the advantage of our approach. Certainly, more advanced encoder-decode networks can be investigated in our future work to realize image mapping, and we believe this could further improve the performance of our approach.
>
> **5. Data augmentation used.**
>
> We use the standard data augmentation method as explained in Sec. A.1 for all the data sets and backbones. The same setting is widely used by the methods in comparison, which include [4, 9, 10, 12, 14, 22, 31, 35, 47, 48] (references in our paper) except those in [7, 34, 49] which use the augmentation designed by themselves. In addition, the data augmentation method used by some previous work [13, 30, 39] is not explicitly described in their papers and no publicly released code is available either. Following the suggestion, the data augmentation information for all the methods in our comparison has been clearly listed in Tables 1, 2, and 3.
>
> **6. Potential negative societal impact.**
>
> This paper focuses on developing a new framework for general domain generalization research. Also, it conducts experiments with the public benchmark data sets that are widely used in this research area. Therefore, we believe that our work in its current form will not have any negative social impact.

---

> > ### Comment · Reviewer_T9VV · 2022-08-08
> > **Thank you for the answers**
> >
> > I thank the authors for the answers. Regarding the answer 1, I agree that the framework is able to remove the domain-specific features from the first channel. However, I still do not think this guarantees that the domain-specific features learnt in the first channel is removed, because there is no reason the second channel does not learn the same feature. But I also understand that the proposed method seems working empirically.

---

### Meta-Review · Area_Chair_zDkR · 2022-08-25

**Recommendation:** Accept
**Confidence:** Certain

**Metareview:**

After the author-reviewer discussion, Reviewer 8K1L shows strong support for the paper, and Reviewer MBUN finds most concerns addressed and upgrades the score to Weak accept. Reviewer T9VV has some remaining concerns, but does agree the proposed method seems working empirically. After careful consideration, AC recommends accepting the paper.

**Award:**

No

---

### Decision · Program_Chairs · 2022-09-14

Accept